# Are Wellness Visits a Possible and Effective Cure for the Increasing Cancer Burden in Poland? Example of Women’s Preventive Services in the U.S.

**DOI:** 10.3390/cancers14174296

**Published:** 2022-09-01

**Authors:** Paweł Koczkodaj, Fabian Camacho, George P. Batten, Roger T. Anderson

**Affiliations:** 1Cancer Epidemiology and Primary Prevention Department, Maria Sklodowska-Curie National Research Institute of Oncology, 02-781 Warsaw, Poland; 2Comprehensive Cancer Center, University of Virginia, Charlottesville, VA 22903, USA

**Keywords:** women, cancer prevention, cancer screenings, risk factors, wellness visits, Poland, United States

## Abstract

**Simple Summary:**

Cancer remains one of the leading causes of death in Poland as well as in the United States. Modifiable risk factors, such as tobacco use, obesity, and low utilization of preventive cancer screenings, all contribute to greater cancer incidence and mortality in both countries. However, these risk factors have triggered some mitigating solutions in the U.S not seen in Poland, namely the implementation of standard annual wellness visits fully covered by health insurance. The aim of this study was to determine the level of impact that standardized, preventive wellness visits have had on uptake of cancer prevention measures, namely cancer screenings, among women in a largely rural region of the U.S. where cancer risk is often high. Based on the observation that wellness visits are associated with greater use of cancer screenings and other preventive health behaviors, findings from this study were used to support the inclusion of no-cost wellness visits into proposed strategies for systemic solutions to reduce cancer burden in Poland.

**Abstract:**

Background: Cancer is one of the leading causes of death among Polish women in general, and first in women aged 25–64. Contributing to this cancer burden are modifiable behavioral risk factors, including low utilization of cancer screenings. Poland has an urgent need for new systemic solutions that will decrease cancer burden in the female Polish population. This study examined the United States’ implementation of preventive wellness visits as a viable solution for implementation in Poland. Methods: Health insurance claims data for nearly three million women in five states of the U.S. were examined to identify use of mammograms, colorectal cancer screening, and lung cancer screening. Three subgroups of the cohort were assessed for the probability of receipt of screening associated with type of healthcare visit history (women with wellness visits—W; with wellness visits and related preventive services and screenings—W+P; and control group—C). All multiple comparisons were significant (alpha = 0.05) at *p* < 0.0001, except comparison between subgroups (W vs. P+W) for lung cancer screening. Results: Breast and colorectal cancer screenings had substantially higher participation after W and W+P in comparison with C; moreover, a slight increase after W or P+W was seen for lung cancer as well. Conclusions: Results indicate that wellness visits are an effective tool for increasing cancer screening among women in the U.S. Introduction of a similar solution in Poland could potentially help produce higher screening rates, address cancer prevention needs (not only for secondary cancer prevention), and lower cancer burden.

## 1. Introduction

According to the latest data from the Polish Cancer Registry, more than 171,000 people were diagnosed with cancer in Poland in 2019. Cancer is the second leading cause of deaths in Poland, contributing to 25.7% of all deaths among males and 23.2% among females. Moreover, cancer is a particular health challenge among females aged 25–64 where it is the leading cause of death (31.7% of all deaths in young women and 46.8% in middle-aged women). The most frequent cancer sites in women in 2019 were breast (C50)—19,620 cases (ASW: 55.4); lung (C34)—8,469 (ASW: 40.1); and colon (C18)—5,043 (ASW: 24.3). Considering mortality data, the most frequent causes of cancer deaths among Polish women were lung cancer—8,205 cases (ASW: 17.6); breast cancer—6,951 (ASW: 15.0); and colon cancer—3,537 (ASW: 6.3) [1]. Notably, a concerning trend of increasing lung cancer mortality among women has already been observed for many years in Poland (unlike a clear decrease among the male population) [2]. The intersection of breast and lung cancer mortality trends occurred in Poland in 2004. Since that time, lung cancer has been the leading cause of cancer deaths among Polish women, despite the fact that breast cancer is more frequent [3]. This phenomenon is clearly connected with smoking behaviors presented by women within the past decades. Currently about one fourth of the Polish adult population smoke cigarettes—32% of men and 20% of women—(current smokers aged 15 years old and older) [4]. Additionally, some studies indicate an increasing smoking prevalence among girls and young women in Poland (15–19 years old) [5] that can be perceived as a highly valid predictor for the future lung cancer burden in the population.

Apart from cigarette smoking, there are several other factors significantly impacting cancer burden among women in Poland. Similar to tobacco consumption, these factors are highly modifiable. Poor diet, overweight and obesity, and physical inactivity are evidence-based factors with a significant meaning for cancer burden in Polish women. It has been estimated that about 24% of all deaths in Poland are connected with unhealthy diet [6]. In this context it should be also mentioned that about 3.5% of all cancer cases among women are related to alcohol consumption [7]. Within the last decade, alcohol intake in Poland has increased nearly every year, reaching the average level of consumption of 9.6 L of pure alcohol per capita in 2020 [8]. Furthermore, available data show exceptionally low percentages of adequate physical activity levels among Polish women (aged 15 years old and older)—only 16% met minimal recommendations (moderate-intensity exercise ≥ 30 min/day, 5 days a week (150 min/week) or 5 days a week of intense exercise (75 min/week)) for physical activity intensity and frequency in their leisure time, and only 18% of them in leisure time that also included transportation to work [9]. These phenomena are clearly reflected in increasing rates of overweight and obesity. According to 2019 Eurostat data, nearly 50% of adult women (18+) in Poland were overweight and nearly 20% of them were obese [10].

Moreover, participation in cancer screenings is persistently one of the biggest cancer prevention concerns in Polish women. As of 1 February 2022, only 33.21% of eligible women (aged 50–69) undergo regular mammography. Even lower percentages are observed in the case of cytology—just 12.1% of eligible women (aged 25–59) performed this examination at the beginning of 2022 (data retrieved from the National Health Fund reflects participation rate solely for those reimbursed for NHF cytologies) [11]. While screening rates were (surprisingly) not much higher before the COVID-19 pandemic, total colonoscopy percentages for eligible men and women (age group 55–64) were as low as 5% in each month for May through July of 2020 [12].

Considering the aforementioned data on risk factors exposure, there is, undoubtedly, an urgent need to seek new, systemic solutions that will decrease cancer burden in the female Polish population. In light of the fact that participation rates in cancer screenings are considered low or even very low in Poland, significant improvement in cancer deaths can possibly be achieved much faster than in primary cancer prevention cases (e.g., decreasing smoking prevalence demands much more time taking into account the strong addictive character of nicotine).

In this context, a very promising tool—no-cost annual wellness visits (W) (Appendix A) [13]—were developed and introduced in the United States within the health care reform signed into law by President Barack Obama in 2010 [14]. The assumptions of this solution are based on regular annual health checks including identification of and exposure to cancer risk factors in the patient’s daily life. Moreover, during a wellness visit health professionals also provide the essential information about cancer screenings. By providing these checkups that focus on maintaining wellness and disease prevention, rather than on curing or remedying illness—and at no cost to (insured) patients—the goal of this legislation was to increase healthcare by improving preventive care behaviors. Indeed, one of the first national studies on the use of annual wellness visits found that those who took advantage of these visits received a total number of general preventive services that was 62% higher than those who forwent wellness visits [15]. Therefore, the aim of this study was to investigate the effectiveness of wellness visits in terms of chosen cancer screenings participation rates among U.S. women between 2016 and 2017 who resided in five U.S. states. Furthermore, we would like to discuss and consider the potential adaptation of a similar system of using wellness visits to improve cancer prevention health behaviors among women in Poland.

## 2. Materials and Methods

The study population of almost three million women (Table 1) included all women aged 65–79 enrolled in Fee for Service (FFS) Medicare who resided within the geographic regions of Appalachia, within five U.S. states: Virginia (VA), West Virginia (WV), Kentucky (KY), Pennsylvania (PA), and Ohio (OH) (Figure 1) and who received a code identifying screening or preventive services (Appendix B).

The Appalachian region is of importance because it comprises both large rural tracts—which are medically underserved, have generally low access to preventive services, and have historically high cancer burden, and also includes urban and metropolitan centers with better access and health supply. This representation of both rural and urban centers is generally similar to the geographic makeup of Poland [16]. With low access to physicians and specialty care services, the Appalachian region also has increased cancer burden compared to the rest of the country. Importantly, while the U.S. at large has seen favorable declines in cancer incidence and mortality rates over the past two decades, rates across the Appalachian region have remained comparatively high and are now among the highest in the country [17], which may be considered as another analogy to Poland in the context of other European Union (EU) member states (particularly regarding cancer mortality). Moreover, according to 2019 Global Burden of Disease data [18], West Virginia and Kentucky are the only two states with exceptionally high disability-adjusted life years (DALY) indicator levels attributable to tobacco use (17·5% to <20%). Similarly, Poland is one of the few EU states (Bulgaria, Croatia, Greece, Hungary, and Poland) with tobacco attributable DALY at the same high range, where the role of tobacco consumption—particularly in female populations—can be considered as a crucial one [2].

Data for this study were accessed under a data use agreement (DUA) between the University of Virginia (UVA) and the Center for Medicare and Medicaid Services (CMS), and were de-identified to protect individual privacy. This study was conducted at UVA under an approved Institutional Review Board (IRB) protocol.

Medicare claims data were examined from 1 January 2016 to 31 December 2017 (2 years) to identify use of three key preventive services: mammograms, colorectal cancer screening (including colonoscopy and fecal blood test), and lung cancer screening (low-dose computed tomography—LDCT). Following Camacho et al. [15], to parse the effect of the Annual Wellness Visits on follow-up screening from alternative health care visit patterns which may have influenced screening, and which may have occurred sufficiently close to and prior to the wellness visit, we performed a stratified analysis with a 90-day look-back period to the wellness visit to record presence of preventive services (P) prior to a wellness visit. The investigated cohort was therefore divided into the following subgroups:

Women with a first or repeated wellness visit without any preventive services within previous 90 days of the index visit. These were classified as Group ‘W’ or those who had an index period wellness visit and had no recent history of prior preventive services.

Women whose index wellness visit occurred within a 90-day period after receipt of preventive services. These were classified as Group P+W or those who already had received preventive services and made a subsequent wellness visit.

A control group (C) of women who did not have a wellness visit within a random reference date and who had not received a study-observed preventive service.

All women included had to have at least one year enrollment in Medicare FFS (before and after W or the random reference date for C) and no additional W during the follow-up (365-day period). For all three screening services of interest, we assessed the probability of subsequent receipt of screening based on healthcare visit group.

For all multiple comparisons, the threshold for statistical significance was set to (alpha = 0.05) at *p* < 0.0001, except for the comparison between subgroups no. 1 and no. 2 for lung cancer screening. For the latter, because approved lung cancer screening programs for patients with eligible smoking histories had just begun to be implemented in our study region within the timeframe of our study and were not offered in many county locations [19], the outcomes for use of lung cancer screening were calculated for all eligible beneficiaries located in a sample of 20 counties selected as having the most lung cancer screening claims (Figure 2 and Table 1). Moreover, because we lacked data on smoking status, the effects of wellness visits was estimated at the Medicare insured population level among all female beneficiaries; thus, we could not estimate effects specifically for smokers.

In order to avoid bias connected with differences in the rates caused by other independent factors, we used inverse probability weights for multiple groups (IPTW). We applied adapted multinomial logistic regression for membership probabilities used further for the weights. In our model we used covariates listed in Figure 3. We also identified a number of confounders which can be considered as potential predictors of results such as: National County Economic Distress (county economic status) from the Appalachian Regional Commission [20]; National Center for Health Statistics (NCHS) rural urban status from the Centers for Disease Control and Prevention (CDC) [21]; race (black, white, other); age at time of reference; Charlson comorbidity index (based on claims ICD-9/10 diagnosis codes with categorized scores ranging from 0–4+) [22]; and reference date.

## 3. Results

After considering all of the criteria, a total of 2,869,072 insured women were selected for study. Of these, 27% (*n* = 788,779) had made a Medicare reimbursed wellness visit (W) within the one-year observation period; 4.5% (*n* = 35,847) of the women used wellness visits (W) exclusively, and 19% (*n* = 149,329) had W and one or more related preventive services or screenings (P+W) within 90-days period before W. The study control group (C) consisted of 8% (*n* = 168,516) of women (Table 1). For the separate lung cancer screening sample, 93,325 women were identified with residence among the 20 selected study counties. Of these, approximately 10% (*n* = 9,467) were classified as W, 43% (*n* = 40,119) classified as W-P, and 47% (*n* = 43,739) were classified as C.

A great majority of the women in this sample came from counties with a transitional economic status (approximately 68%) (according to Appalachian Regional Commission information, “transitional counties are those transitioning between strong and weak economies. They make up the largest economic status designation. Transitional counties rank between the worst 25 percent and the best 25 percent of the nation’s counties”) [20] and lived in medium-sized metropolitan areas (about 26%) (according to the NCHS Urban-Rural Classification Scheme for Counties, “medium metro counties are counties in metropolitan statistical areas of 250,000 to 999,999 population”) [21]. Moreover, white women comprised 90% of the sample, having the highest level of representation. The largest part of the cohort represented those aged 70–74 (37%); however, the youngest age group (65–69) was smaller by only about 2%. A vast majority of women—almost 59% of them—had no Charlson index-defined comorbidities, while at the same time the most frequent comorbidity was diabetes (19%), followed by chronic obstructive pulmonary disease (COPD) (15%). Additionally, the highest number of visits had reference dates in the first investigated semester (27%) and the lowest in the second semester at 22% (Figure 4). Further, relatively high differences in percentages were observed in some cases between W and W+P, particularly for absence of Charlson index—58% vs. 39%, respectively, but also for women with index at level of 2,3—14% vs. 23%, and 4+—approximately 4% vs. 11%. Moreover, considerable differences were investigated between W and W+P for comorbidities such as: PVD—6% vs. 12%; CVD—5% vs. 10%; COPD—13% vs. 24%; renal disease—6% vs. 10%; and for diabetes 19% vs. 28% (Figure 3).

Finally, we investigated changes in cancer screening participation among women after W, P+W, and C during 1-year follow-up. Due to the large sample size, almost all comparisons were highly significant (the confidence intervals illustrated as error bars), except comparison between W and P+W for lung cancer. In the case of breast and colorectal cancer screenings, C was characterized by substantially lower participation. Nevertheless, in lung cancer cases we also observed a slight increase after W or P+W (0.5% and 0.4% increases, respectively). We also discovered that W and P+W were particularly effective in increasing breast cancer screening participation—for W an increase of about 25%, for P+W more than 35%. In the case of colorectal cancer screening differences between C and after W or P+W, these are lower—9.2% and 12.9%, respectively; however, here the difference between W and P+W is much lower (3.7%), unlike that for breast cancer screening (10.5%) (Figure 5).

As mentioned in Materials and Methods (above), in order to more precisely present the effect of W and P+W on lung cancer screening participation, we decided to analyze the 20 counties with the most lung cancer screening claims. In this case, we received a sample of 9,467 women with W only, 40,119 for W+P, and 43,739 in the control group (Table 1). Contrary to the results presented in Figure 5, we can assume that in the case of lung cancer screening there could be a slight difference in effectiveness between W and P+W in favor of the W alone (Figure 6). Moreover, C is characterized by the lowest participation—similar to breast and colorectal cancer screenings. Analyzed data indicate also the lowest efficiency of W and P+W in increasing lung cancer screening participation among women (C vs. W: 0.50% increase; C vs. P+W: 0.34% increase) and warrant further investigation in future studies.

## 4. Discussion

In our study, we investigated a significant increase in the effectiveness of W in terms of cancer screening participation among women, particularly among breast and colorectal cancer screenings—about a 25% increase and 9% increase, respectively, compared to control groups. These results, from the first U.S. study conducted on this topic in a largely underserved rural region, are in line with the study conducted by Camacho et al. using a national sample [15]. In the latter, the authors proved that W increased screenings rates substantially; however, the sample in this previous study had quite different parameters (e.g., different age groups and analyzed period, it included both sexes, and it excluded lung cancer screening). Moreover, other studies by Galvin et al. [23] and Tong et al. [24] showed improved utilization of preventive services as a result of W, as well as elevated involvement of patients in issues connected with prevention generally.

In Poland, the only tool so far that could be considered as similar (in a very general sense) was developed and launched on 1 July 2021. This nationwide program, “Prophylaxis 40 Plus”, is specific to the general population for persons 40 years of age and older. Led by the Ministry of Health (MoH) and the National Health Fund (NHF) in Poland, the main aim of this initiative was an assessment of the organization and effectiveness of preventive diagnostics for most common health problems, including cancers. Practically, the program is based on an electronic health survey which patients fill out on their own. As a result, a system determines whether to generate a referral for examination. The survey also contains a dedicated focus on cancer screenings, though it has solely passive (informative) detail [25]. Until now, there is no official data on the effectiveness of the program (according to the media statements of experts, so far about 1% out of 20 mln eligible population benefited from the program “Prophylaxis 40 Plus”; for comparison, in the U.S. during first year of W functioning (2011), W use was at the level of 7.5% [26]; however, considering the data from the NHF (Table 2) [11] for the analogous period, we can assume a very limited impact of the proposed solutions on cancer screening participation among Polish women. The percentages are even lower than in July 2021 when the aforementioned program had started. Nevertheless, “Prophylaxis 40 Plus” is a pilot program slated to finish in December 2022, creating an opportunity to introduce some changes—similar to W—but at the same time generates risk of no follow-up.

Importantly, in our analysis, isolating W from P+W we were able to show that the impact of W is not due to confounding from a predisposition to use of preventive services. The fact that P+W had a stronger impact on subsequent screening uptake supports our assumption that persons who are already receptive to prevention health services, but who have not completed recommended screenings, can be readily motivated to complete recommended screenings. Thus, even those who already use some preventive services appear to benefit from annual wellness visits, for example, connected with different chronic diseases (e.g., diabetes). Undoubtedly, this phenomenon may also be connected directly with much better patient knowledge of prevention provided by medical professionals during frequent contacts with the healthcare sector and the dedicated time allocated to prevention in the well visit format. In this context, lack of direct contact with medical staff may be considered as one of important factors that may explain a very low impact of “Prophylaxis 40 Plus” on cancer screenings coverage among Polish women. Provision of written information alone seems to be highly insufficient, especially with the paper screening invitations sent in Poland in the years 2009–2015. Percentages of women who participated in screenings as a result of receiving written invitation were far from satisfactory (which was also one of the reasons why shipping was discontinued after 2015) (Table 3) [27]. By contrast, the informative and broad educational character of the W (including preparation of a screenings schedule for the next years during face-to-face meetings) is often considered one of the most beneficial features of this solution [24].

We hypothesize that the higher percentages of P+W vs. W screened might have a connection with better health education and the overall higher education level of women who had other related preventive services. Chien et al. indicated a strong positive correlation between level of education and the willingness to participate in health screenings for chronic diseases [28]. This factor can be also a possible reference for our study, as a vast majority of investigated women lived in metropolitan areas characterized generally by higher levels of education [29]. In the case of a Polish program, a short informational campaign turned out to be unsatisfactory. Moreover, for years there was no coordinated lifelong health education in schools that targeted health promotion and, particularly, cancer prevention. In 2019, the Ombudsman for Patients’ Rights proposed the introduction of the subject “Health knowledge” into the Polish education system; however, this has not happened so far [30].

Finally, we assessed the impact of W and P+W on lung cancer screening participation among women. To the best of our knowledge, our study is the first to investigate this relationship. Similar to breast and colorectal cancer screenings, we showed a positive influence of W and P+W on participation rates. However, in contrast to the previously described screenings, in the case of lung cancer (20 counties sample—Figure 5) the percentage for W was higher than P+W and that may be directly connected with characteristics of this particular group. Smokers less often participate in screenings and visit a doctor and, thus, without data on smoking status, we are unable to determine if the smoking rates between P+W and W were substantially different. However, in the 20 counties examined in which lung cancer screening services were known to be available during the study period, having completed a wellness visit was clearly associated with better uptake, which is overall consistent with the findings of benefit. Meanwhile, the latest recommendations of the U.S. Preventive Services Task Force (2021) assume annual lung cancer screenings among smokers aged 50–80 years old [31,32].

In Poland, there have been some attempts to launch a Nationwide Lung Cancer Screening Program [33,34]. Until now, however, there is no solution analogous to the other three prevention screenings programs funded by the NHF (breast, cervical, and colorectal cancer screenings). In the face of an exceptionally high lung cancer burden, as well as constantly increasing lung cancer mortality, among Polish women [1,2], W could be an excellent, effective tool supporting smoking cessation, but also lung cancer screening participation in the future.

### Limitations of the Study

Our applied method (IPTW) does not necessarily include all possible confounders that may contribute to some spurious associations. Moreover, the low number of women who participated in lung cancer screening limits inference. In addition, while breast and colorectal cancer screening is recommended for all women based on age, referral for lung cancer screening is only applicable to those with a specific history of smoking (pack-years). We lacked data on smoking status and pack-years, and our results assume that visit type (P, P+W, or C) is unrelated to underlying smoking status or history. The validity of this assumption is untested. However, despite these limitations, the results were largely consistent across screening service, showing a benefit of insurance-covered wellness visits on service uptake.

Due to certain differences (e.g., different inclusion cancer screening criteria in the U.S. and in Poland, lack of an organized lung cancer screening program in Poland, different health and socio-economic factors, etc.), obtained results cannot be literally transferred to the Polish female population, but they are nevertheless valuable indicators that constitute a basis for further public health interventions with the goal of increasing cancer screening participation rates among Polish women.

## 5. Conclusions

Annual wellness visits (W) are an effective public health tool, increasing cancer screening participation rates among women.

Introduction of W, or its elements, could potentially promote higher screening participation rates and, in the most long-term perspective, lower cancer burden among Polish women. Moreover, W could also address urgent primary cancer prevention needs (for example, those connected with mentioned physical inactivity, obesity, or alcohol consumption among women in Poland). The introduction of annual wellness visits could be possibly performed within the forthcoming changes in the program “Prophylaxis 40 Plus”.

The main challenges to introducing these solutions, which are connected to financing and organization, seem possible to overcome with current resources (e.g., involvement of other-than-medical-doctors health specialists—for example, pharmacists or public health specialists; partial use of telemedicine tools; financing of an annual visit within a capitation rate in primary health care).

Improvement in screening participation rates demands complex and complementary solutions. In order to achieve the full impact of wellness visits in the future, there is a dire need to introduce health education within the education system in Poland.

As our study was performed in an American population and the investigated group was exposed to many other factors, it is hard to predict the final size of potential influence of annual wellness visits on cancer screening rates in Poland. However, any resulting impact will undoubtedly be a positive change. Therefore, once again considering cancer epidemiological data, as well as risk factor exposure for Polish women, annual wellness visits should be definitely considered as a potential tool to introduce.

## Figures and Tables

**Figure 1 cancers-14-04296-f001:**
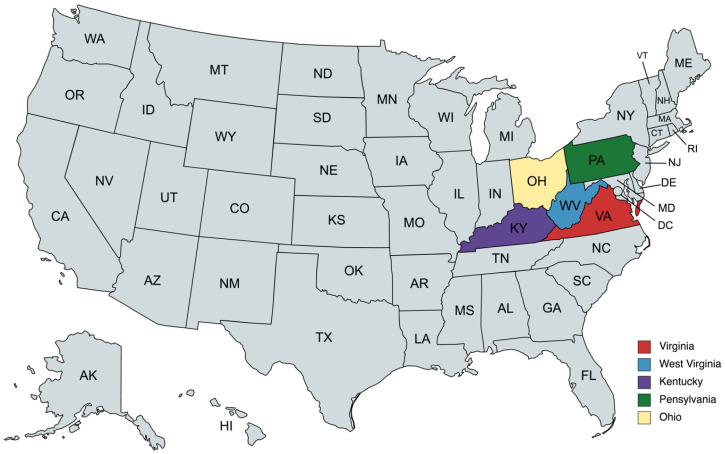
Location of the analyzed U.S. women’s population (map created with mapchart.net).

**Figure 2 cancers-14-04296-f002:**
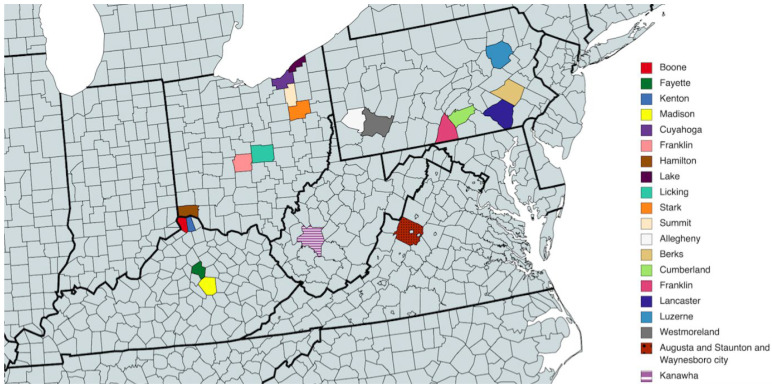
Counties included in the analysis on lung cancer screening (map created with mapchart.net).

**Figure 3 cancers-14-04296-f003:**
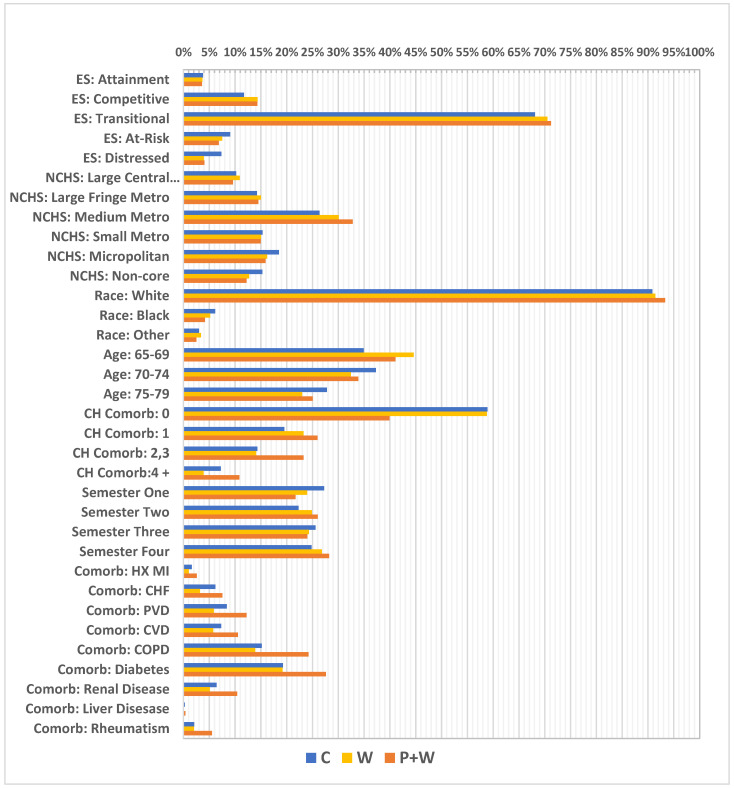
Unweighted covariate distributions. ES—economic status; NCHS—National Center for Health Statistics; Metro—metropolitan; CH Comorb—Charlson comorbidity index; HX MI—history of myocardial infraction; CHF—chronic heart failure; PVD—peripheral vascular disease; CVD—cardiovascular disease; COPD—chronic obstructive pulmonary disease.

**Figure 4 cancers-14-04296-f004:**
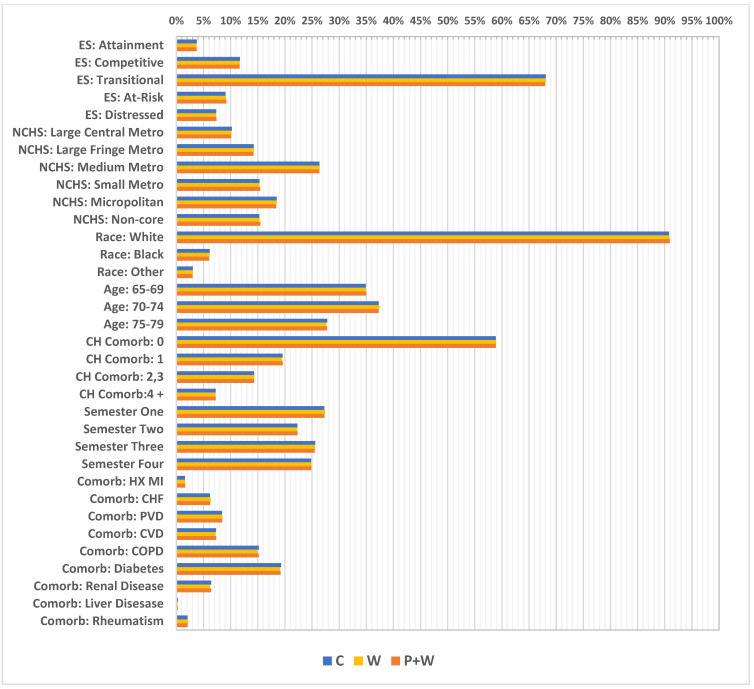
Weighted covariate distributions (after weighting, the sample rates between subgroups became similar). ES—economic status; NCHS—National Center for Health Statistics; Metro—metropolitan; CH Comorb—Charlson comorbidity score; HX MI—history of myocardial infraction; CHF—chronic heart failure; PVD—peripheral vascular disease; CVD—cardiovascular disease; COPD—chronic obstructive pulmonary disease.

**Figure 5 cancers-14-04296-f005:**
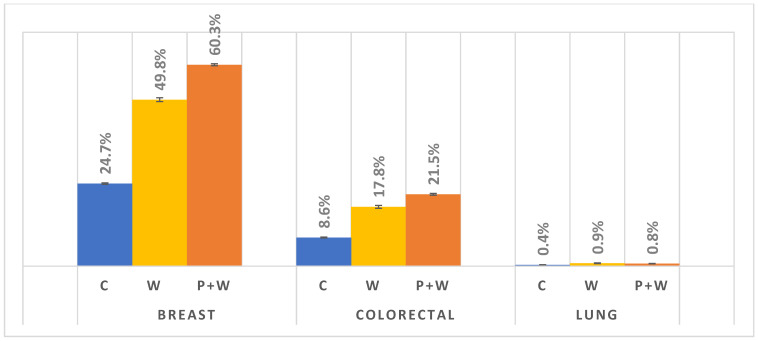
Cancer screenings participation during 1-year follow-up.

**Figure 6 cancers-14-04296-f006:**
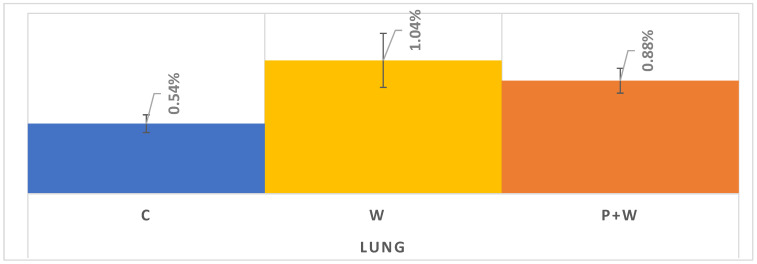
Lung cancer screening during 1-year follow up in selected 20 counties with access to LDCT.

**Table 1 cancers-14-04296-t001:** Primary cohort and extracted sample characteristics.

Total Number of Beneficiaries in the Years 2014–2018: 2,869,072			
Beneficiaries with Wellness Visits (W): 788,779		Beneficiaries without Wellness Visits: 2,080,293	
Characteristic of beneficiares	Number	Characteristic of beneficiaries	Number
Beneficiaries enrolled in FFS Medicare +/− 1 year from a candidate’s reference day	532,201	Beneficiaries enrolled in FFS Medicare +/− 1 year during time window 2016–2017	715,210
Beneficiaries with no additional W during a reference time window 2016–2017	514,321	Beneficiaries without 90 days lookback for related preventive services or screenings (P) before reference	350,854
Beneficiaries with reference days only after 1 January 2016	302,397		
Beneficiaries aged 65–79 at reference day	229,792	Beneficiaries aged 65–79 at reference day	197,360
Beneficiaries restricted to 5-states study region at reference	185,176		
**↓** **FINAL SAMPLE** **↓**			
W only	35,847	Control group (C) (after restricting beneficiaries to 5-state study region at reference)	168,516
Previous related preventive services or screenings (P) + W	149,329		
**LUNG CANCER SAMPLE (20 CHOSEN COUNTIES)**			
W only	9467	Control group (C)	43,739
Previous related preventive services or screenings (P) + W	40,119		

**Table 2 cancers-14-04296-t002:** Cancer screenings coverage in eligible population of women in Poland. Period: July–December 2021; January–February 2022. Percentages reported by the NHF on the first day of each month.

Year	2021						2022	
Month	Jul	Aug	Sept	Oct	Nov	Dec	Jan	Feb
Cytology (%)	12.98	12.90	12.82	12.77	12.65	12.56	12.60	12.10
Mammography (%)	33.98	34.25	34.52	34.73	34.90	35.14	35.31	33.21

**Table 3 cancers-14-04296-t003:** Effectiveness of written invitations shipment in Poland in the years 2009–2015.

Year	Number of Sent Invitations		% of Women Who Conducted Screening after Receiving an Invitation	
	Mammography	Cytology	Mammography	Cytology
2009	1,130,217	1,595,311	26.83	13.18
2010	2,419,462	3,202,921	26.59	11.73
2011	2,536,771	3,357,113	29.68	12.70
2012	2,690,028	3,413,678	26.07	9.68
2013	2,603,090	3,220,579	24.04	9.87
2014	2,749,920	3,348,593	21.42	9.20
2015	2,970,912	3,432,764	21.13	9.52

## Data Availability

Data for this study were purchased and accessed under a data use agreement (DUA) between the University of Virginia (UVA) and the Center for Medicare and Medicaid Services (CMS), and were deidentified to protect individual privacy. This study was conducted at UVA under an approved Institutional Review Board (IRB) protocol. Claims data at this detailed level are not provided free to the public by CMS. The study team may be contacted for names of specific data files and diagnostic codes used to complete these analyses.

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
