# Peer review of "Are Wellness Visits a Possible and Effective Cure for the Increasing Cancer Burden in Poland? Example of Women’s Preventive Services in the U.S."

_cancers, 2022, doi:10.3390/cancers14174296_

Round 1

Reviewer 1 Report

This paper makes a novel and important contribution to the literature. The focus on a rural Appalachian population and impact of wellness visits is innovative and comparison with lessons for Poland (and potentially other countries) is intriguing. 

Overall the results are clearly presented and discussed, but have some minor suggestions for improving the paper: 

Pages 2-3: The argument for focusing on screening (versus primary prevention) could be made clearer here. Much of the Introduction focuses on causes and primary prevention of cancers among women, with relatively brief background on screening (third paragraph). It would be helpful to know more about the cancer screening context. 

Page 2 Lines 57-62: It might be helpful to explain here why lung cancer has lower survival, especially with later discussion re screening. The incidence of lung cancer is tied to smoking, the lower survival is due to limited options for early detection and intervention (compared with breast cancer). It would also be helpful here to provide prevalence for smoking among both women and men in Poland for comparison. 

Page 2 lines 69-77: It would be helpful to provide comparable figures for men as well to understand how the impact of these risk factors may be different for women. 

Page 3 Lines 91-93: This is a poor example for making the point about priority of screening (versus primary prevention). The impact of lung cancer screening on reducing mortality is very small compared with the impact of smoking cessation and prevention. It is true that the mortality impact of screening in older smokers will be seen sooner than the impact of preventing smoking among youth, for example, but this is because of the long lag time between smoking uptake and development of lung cancer (not directly because of the addictive nature of nicotine, as your text suggests). Can you find another example to make this point?  

Page 5: Can you state what the eligibility criteria are for lung cancer screening in Poland? 

Figures 3 and 4: I was going to ask why you didn't include smoking prevalence in these figures also across the 3 groups (along with other factors) but I see you note in the Discussion this information was not available. It would be useful to explain this earlier or in Methods. 

Page 11 lines 335-336: Agree that Wellness Visits could be a good opportunity for both tobacco cessation counseling and also referral for screening!

Author Response

Dear Reviewer,

Please find the attached answers to your comments and suggestions. 

Sincerely yours,

Authors

Reviewer 2 Report

Dear Authors,

thank you for the possibility to read and review your research.

The study deals with a very important issue: increasing the possibility and availability of prophylactic methods aimed at reducing the incidence of cancer in Polish population. It is known that the health services all over the world focus on the treatment of cancer. All of the concepts that take into account the possibility of earlier detection of malignancy and even avoiding it, are worth presenting and discussing.

This article is well designed, the methods are described in a clear way. The results are presented well. The discussion is interesting.

I think that it would be worth supplementing the information on preventive programs in Poland, which, although not in a similar way, but are carried out, such as the program of prevention of tobacco diseases, prevention of breast cancer or colon cancer. Authors mention just Prophylaxis 40 plus. Another issue is the problem that patients do not like to participate to these programs.

However, these are general remarks.

In conclusion, I recommend the publication of this work in its present form.

Author Response

Dear Reviewer,

Please find the attached answers to all your comments.

Sincerely,

Authors
